# Uni-DocRobust: Universal Plug-and-Play Robustness Enhancement for Multi-modal LLMs via Feature Restoration

**Yuxuan Zhou** [1]  **Baole Wei** [2 3]  **Xingjian Hu** [1]  **Haowei Chen** [4]  **Yu Li** [1]  **Xingyue Lin** [1]  **Liangcai Gao** [1]  **Zhi Tang** [1]

## Abstract

Real-world degradations, such as noise, blur, and low resolution, significantly impair the performance of Multi-modal Large Language Models (MLLMs) in document understanding tasks. Despite recent advancements, progress in this field remains stifled by two critical bottlenecks: the scarcity of large-scale, aligned training data necessary for learning robustness, and the lack of transferable restoration solutions across diverse MLLM architectures. To bridge the data gap, we first present DocRobust-VQA, a large-scale dataset explicitly constructed to support robustness training. Comprising 189K aligned clean/corrupted document image pairs and 417K QA pairs, it provides the first substantial corpus for fine-tuning MLLMs to handle varying degradation conditions. Leveraging this data, we propose Uni-DocRobust, a universal plug-and-play framework that decouples restoration capabilities from specific visual encoders. Our method employs a frozen Universal Restoration Core pretrained in a canonical feature space via multi-teacher distillation, which can be seamlessly integrated into target MLLMs (e.g., Qwen-VL, InternVL) through lightweight Feature Adapters. Extensive experiments demonstrate that Uni-DocRobust significantly enhances robust performance on MLLMs and enables a cost-effective "pre-train once, deploy everywhere" paradigm for robust MLLM deployment.

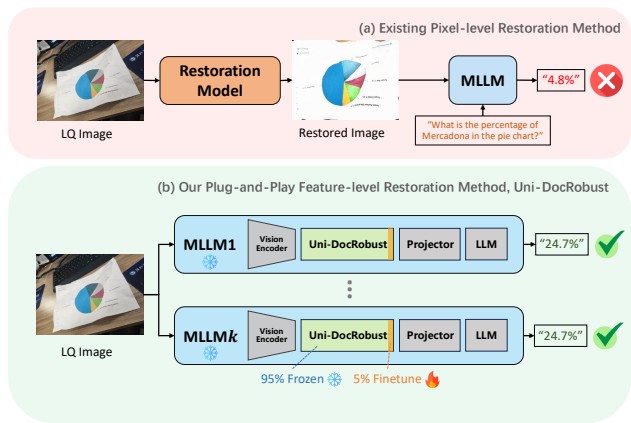

*Figure 1.* Comparison between existing pixel-level restoration method and our plug-and-play feature-level restoration method, Uni-DocRobust.

## 1. Introduction

Documents are primary carriers of knowledge and information, and their practical parsing and comprehension are crucial for improving information processing efficiency and enabling digital workflows (Xu et al., 2020a). Document understanding has demonstrated significant value in various fields, including information extraction, text recognition, and knowledge management.

In recent years, Multimodal Large Language Models (MLLMs) have advanced rapidly, and document understanding has emerged as a key application (Liu et al., 2024; Luo et al., 2024; Hu et al., 2024; Ye et al., 2023). Capabilities such as text recognition and visual document question answering have gradually become major optimization objectives (Liao et al., 2023; Blecher et al., 2023; Wang et al., 2024b). The continuous progress in these models provides robust technical support for document understanding, enabling cross-modal information fusion, and promoting a shift from traditional single-modal processing to comprehensive multimodal interpretation. However, in practical scenarios, document images often suffer from noise, blur, low resolution, and other degradations that lead to significant recognition and comprehension errors (Das et al., 2019;

---

[1]Wangxuan Institute of Computer Technology, Peking University, Beijing, China [2]Zhongguancun Academy, Beijing, China [3]Zhongguancun Institute of Artificial Intelligence, Beijing, China [4]School of Electronic Engineering and Computer Science, Peking University, Beijing, China. Correspondence to: Baole Wei <weibaole@zgci.ac.cn>, Liangcai Gao <gaoliangcai@pku.edu.cn>.

*Proceedings of the 43$^{rd}$ International Conference on Machine Learning*, Seoul, South Korea. PMLR 306, 2026. Copyright 2026 by the author(s).

Zhang et al., 2024a; Lin et al., 2020).

Although recent benchmarks such as R-bench (Li et al., 2024a) and WildDoc (Wang et al., 2025) have been introduced to evaluate the robustness of large models under low-quality visual conditions, there remains a significant gap in methods specifically designed for enhancing the robustness of MLLMs via low-quality image restoration. We attribute this gap to two main factors. On the one hand, while benchmarks for evaluation are becoming available, MLLMs typically require large-scale datasets for effective fine-tuning, and existing low-quality document image datasets are far from sufficient in scale or diversity. On the other hand, as shown in Fig. 1, existing methods for low-quality document image restoration (Zhang et al., 2024a; Souibgui & Kessentini, 2020) typically focus on pixel-level recovery. Such dense supervision may lead to overfitting and, coupled with repetitive visual feature extraction and dense pixel prediction, incurs substantial computational overhead—rendering joint optimization with document understanding tasks difficult.

To address these challenges, we present a holistic solution comprising (1) a data foundation and (2) a universal methodological framework. We first construct DocRobust-VQA, a large-scale dataset containing over 189,771 aligned clean-blurry document image pairs and 417,502 question-answer annotation pairs. DocRobust-VQA provides sufficient scale and diversity to support the training of multimodal large language models to gain robustness under degraded visual conditions.

Building upon this data, we propose Uni-DocRobust, a novel universal plug-and-play framework designed to decouple document restoration from specific visual encoders. Uni-DocRobust introduces a decoupled architecture consisting of a Universal Restoration Core and lightweight Feature Adapters. Our method adopts a three-phase strategy: (1) Universal Pre-training: We distill knowledge from multiple heterogeneous visual encoders into the restoration core, enabling it to learn canonical restoration patterns that are agnostic to specific model architectures. (2) Fast Adapter Tuning: To adapt the pre-trained core to a specific target MLLM, we freeze the core and train only lightweight Feature Adapters, focusing on efficiently aligning the canonical features with the target model's specific distribution, ensuring seamless compatibility with minimal computational cost. (3) Plug-and-Play Deployment: Then the adapted module is injected into the target MLLM via a non-intrusive hook mechanism, which allows the MLLM to be upgraded with robustness capabilities instantly, without modifying its original weights or requiring any fine-tuning of the large model itself.

In summary, the main contributions of this work are as follows:

- We introduce **DocRobust-VQA**, a large-scale dataset tailored for VQA on low-quality document images that provides rich diversity and sufficient data volume to facilitate both training and robustness evaluation of multimodal large language models under degraded visual conditions.

- We propose an efficient plug-and-play feature-level restoration method, **Uni-DocRobust**, to restore and supplement lost information in low-quality document images, thereby enhancing the overall robustness of multimodal large language models.

- We conduct a comprehensive comparative analysis of existing multimodal large language models on low-quality document images across various dataset and benchmark. Extensive experiments show that, after training on DocRobust-VQA, our proposed Uni-DocRobust not only enhances the robustness of MLLMs on the synthetic test sets, but also improves their performance on real-world degraded images, offering new insights and methodologies for robustness research in multimodal large language models.

## 2. Related Works

### 2.1. Visual Document Understanding

Visual document understanding (VDU), a key task in cross-modal learning, has evolved through three main stages. Early works (Xu et al., 2020a;b; Huang et al., 2022; Li et al., 2021a;b; Gu et al., 2021; Appalaraju et al., 2021) focused on pretraining models that combine OCR-extracted text with layout features, achieving strong performance in structured document tasks. To address OCR-related limitations, later methods (Kim et al., 2022; Davis et al., 2022; Tang et al., 2023; Lee et al., 2023) adopted end-to-end architectures that directly extract semantics via visual encoders. Recently, multimodal large language models (MLLMs) (Luo et al., 2024; Liu et al., 2024; 2023; Wang et al., 2024a; Lu et al., 2024; Chen et al., 2024; Hu et al., 2024; Li et al., 2024c; Ye et al., 2023) pretrained on massive datasets have set a new paradigm, showing strong zero-shot and instruction-following capabilities. Although robustness under low-quality inputs has gained attention (Li et al., 2024a), and datasets like WildDoc (Wang et al., 2025) simulate real-world conditions, there remains a lack of effective, MLLM-compatible restoration methods for degraded document images.

### 2.2. Methods for Degraded Document Images

Existing methods for handling degraded document images fall into two categories: data quality enhancement and model robustness improvement.

Data enhancement methods typically focus on image restoration, either targeting specific degradations (Das et al., 2019; Lin et al., 2020; Wang et al., 2022; Yang et al., 2024; Zhang et al., 2022; 2023a;b) or using unified architectures (Souibgui & Kessentini, 2020; Souibgui et al., 2023; Yang et al., 2023) that still require task-specific training and inference. Recent work like DocRes (Zhang et al., 2024a) supports multiple restorations via visual prompts, but such pixel-level methods lack semantic-level optimization and are inefficient for downstream tasks.

Robustness-oriented methods improve model tolerance via training strategies, including masked pretraining (Lyu et al., 2022), contrastive learning (Yang et al., 2022; Guan et al., 2023), and degradation simulation (Wei et al., 2024). DoCo (Li et al., 2024b) enhances visual encoding for dense text. However, these methods often rely on modifying pretraining objectives, limiting compatibility with existing MLLMs.

To overcome this, we propose the Uni-DocRobust which restores feature-level quality without altering the backbone model, significantly boosting robustness for pretrained MLLMs under low-quality document conditions.

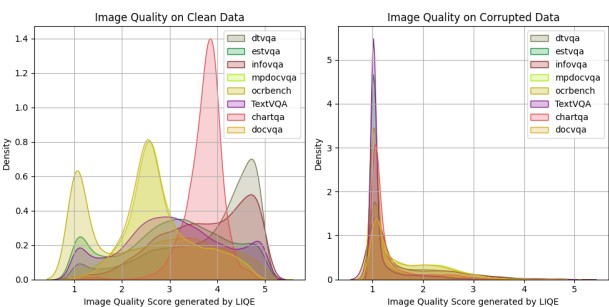

*Figure 2.* Image quality score on clean/corrupted image datasets, generated by LIQE(Zhang et al., 2023c)

## 3. Dataset Construction

To enhance the robustness of multimodal large vision-language models on low-quality document images, we construct a DocRobust-VQA, composed of paired clear document images and their corresponding corrupted versions.

### 3.1. Clean Data Collection

The quality and quantity of clear images in the training set form the foundation of the dataset and are critical for enabling the multimodal model to learn robustness. In selecting clear document images, we considered the following factors: (1) Domain Diversity: The model should generalize across varied image types (e.g., office documents, receipts, charts, scene texts, and text line crops), therefore the training set incorporates diverse sources. (2) Scale Diversity:

Perturbations affect images differently depending on dimensions and text sizes. To ensure robustness, the training set includes both large-format documents with small text and smaller crops or scene texts with larger fonts. (3) Clarity: While real data may contain degraded samples, paired training requires fully legible images to provide high-quality ground truth for restoration. Hence, our clean data ensure superior quality compared to corrupted inputs, as shown in 2.

Based on these considerations, we integrate and filter clear images with high-quality question-answer annotations from eight datasets, including ChartQA (Masry et al., 2022), DT-VQA (Zhang et al., 2024b), EST-VQA (Wang et al., 2020), Single-page DocVQA (Mathew et al., 2021), Multi-page DocVQA (Tito et al., 2023), InfographicVQA (Mathew et al., 2022), TextVQA (Singh et al., 2019), and OCR-Bench_v2 (Fu et al., 2024).

### 3.2. Corrupted Data Construction

Corrupted images are not only used to train the restoration model together with clear images but also serve as fine-tuning data for the multimodal model's SFT, using the question-answer annotations from the clear images. Moreover, when evaluating the multimodal model's low-quality image understanding ability, the benchmark is constructed from these corrupted images. Thus, the method for generating corrupted images is doubly important for enhancing and assessing the model's capabilities.

Based on the clear images collected in the previous section, we generate corrupted images. Specifically, we classify corruptions into five categories according to their visual effects: (1) Luminance, (2) Distortion, (3) Blurriness, (4) Noise and (5) Compression. For a given clear image, we randomly select $k$ categories from these five, then randomly choose one specific corruption from each selected category, and finally apply the $k$ chosen corruptions sequentially to form the corrupted image. It is noteworthy that these five categories of corruption have a specific sequential order when applied to an image. We observed that some corruption effects can overlap or override others (e.g., Gaussian blur may obscure noise), so the order is fixed as listed above. Additionally, for each corruption, the strength is adjusted based on the image size (e.g., the size of the Gaussian kernel, the radius for flexible distortion, etc.).

## 4. Method

Current state-of-the-art multimodal large language models (Wang et al., 2024a; Chen et al., 2024; Lu et al., 2024) typically consist of four components: a Foundation LLM, a Text Encoder, a Visual Encoder, and a Projector MLP. The Foundation LLM, usually built on an existing language

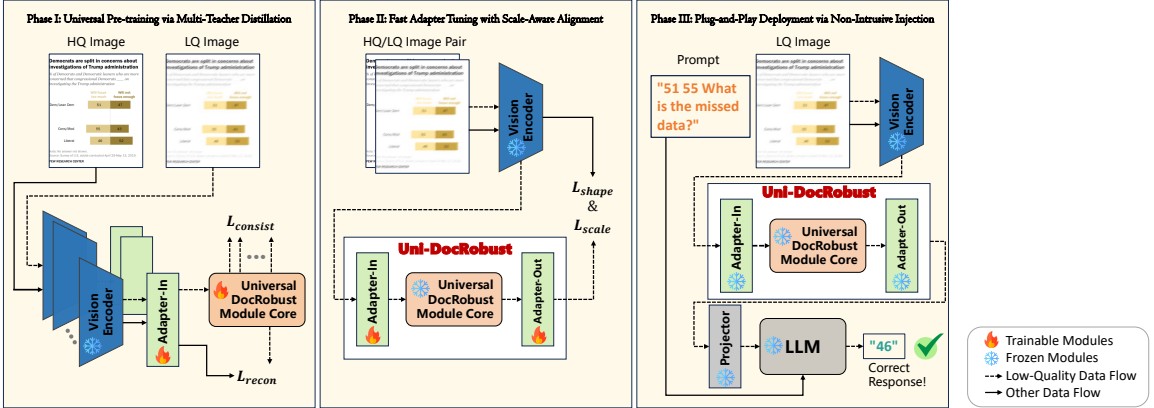

*Figure 3.* The training and inference phases of Uni-DocRobust.

model and trained in an autoregressive manner, is the primary source of the model's comprehension and reasoning. The Text Encoder maps input text into token sequences via an embedding layer and a tokenizer. The Visual Encoder converts input images into token sequences, typically implemented using a Vision Transformer (ViT) (Dosovitskiy et al., 2021). The Projector MLP then maps these encoded visual tokens into a feature space aligned with the Foundation LLM, enabling effective multimodal reasoning between images and text.

However, when processing LQ images, the Visual Encoder or the Projector MLP cannot effectively recover the information lost due to image degradation, leading to misinterpretation by the multimodal model and reduced robustness. To address this, we propose Uni-DocRobust, a plug-and-play framework that decouples the restoration capability from specific visual backbones.

### 4.1. Overall Procedure

As illustrated in Fig. 3, Uni-DocRobust decomposes the robust understanding task into three orthogonal stages: (1) Universal Semantic Restoration, which recovers the lost information in a model-agnostic canonical space, (2) Feature Alignment, which acts as a bridge between specific visual encoders and the restoration core, and (3) Plug-and-Play Deployment, which adopts a Non-Intrusive Injection strategy to enhance the robustness of MLLMs. Formally, our framework consists of a frozen Universal Restoration Core (U-DRM) ($\Phi_{core}$) and lightweight Feature Adapters ($\mathcal{A}_{in}, \mathcal{A}_{out}$). The inference flow for a target MLLM with visual encoder $\mathcal{V}$ is defined as:

$$F_{restored} = \mathcal{A}_{out}\left(\Phi_{core}\left(\mathcal{A}_{in}\left(\mathcal{V}(I_{LQ})\right)\right)\right) + \mathcal{V}(I_{LQ}) \quad (1)$$

where $I_{LQ}$ denotes the low-quality input image. This architecture allows us to *pre-train once* on the fundamental

restoration task and *deploy everywhere* via fast adapter tuning.

### 4.2. Universal Decoupled Architecture

**Universal DocRobust Module (UDRM)**. The core module of Uni-DocRobust is a Transformer-based network designed to operate strictly within a Canonical Feature Space $\mathcal{S}_{can} \in \mathbb{R}^{L \times D_{can}}$. It employs Self-Attention layers to capture long-range dependencies, effectively hallucinating and repairing the broken semantic links in degraded documents.

**Feature Adapters**. To bridge the gap between diverse visual encoders with varying dimensions $D_{enc}$ and the canonical space, we introduce lightweight adapters:

- Adapter-In ($\mathcal{A}_{in}$): Projects corrupted input features from $\mathbb{R}^{D_{enc}} \to \mathbb{R}^{D_{can}}$. It compresses heterogeneous features into the standardized canonical form.

- Adapter-Out ($\mathcal{A}_{out}$): Projects restored features from $\mathbb{R}^{D_{can}} \to \mathbb{R}^{D_{enc}}$. It employs a residual connection design to facilitate gradient flow, ensuring the module refines the features by adding restored details to the original signal.

### 4.3. Training and Inference

#### 4.3.1. PHASE I: UNIVERSAL PRE-TRAINING VIA MULTI-TEACHER DISTILLATION

The goal of the first phase is to train the $\Phi_{core}$ to learn generic restoration primitives—how to recover clear text and layout from blur—that are transferable across models. Instead of training on a single vision backbone, we propose a Multi-Teacher Distillation strategy. We utilize a set of $K$ diverse frozen visual encoders $\{\mathcal{V}_k\}_{k=1}^{K}$ (e.g., CLIP, SigLIP, InternViT) as teachers. Given a pair of aligned high-quality (HQ) and low-quality (LQ) images, the model minimizes the

restoration error $\mathcal{L}_{recon}$ in the canonical space simultaneously for all teachers. Meanwhile, to ensure the restoration core captures high-level semantics agnostic to specific encoder architectures, we explicitly define the Cross-Model Consistency Loss ($\mathcal{L}_{consist}$) as the pairwise cosine distance between globally pooled features. Then the loss in phase I can be formulated as:

$$\mathcal{L}_{univ} = \mathcal{L}_{recon} + \lambda \mathcal{L}_{consist} \tag{2}$$

$$\mathcal{L}_{recon} = \sum_{k=1}^{K} \left( \| \Phi_{core}(\mathcal{A}_{in}^k(F_{LQ}^k)) - \mathcal{A}_{in}^k(F_{HQ}^k) \|_2^2 \right) \tag{3}$$

$$\mathcal{L}_{consist} = \frac{2}{K(K-1)} \sum_{1 \le i < j \le K} \left( 1 - \frac{\bar{\mathbf{z}}^i \cdot \bar{\mathbf{z}}^j}{\|\bar{\mathbf{z}}^i\|_2 \|\bar{\mathbf{z}}^j\|_2} \right) \tag{4}$$

where $\bar{\mathbf{z}}^k = \frac{1}{L} \sum_{t=1}^{L} \mathbf{z}_t^k$ denotes the global average pooling of the restored canonical features from the $k$-th encoder. This constraint forces the Universal Core to align feature distributions across heterogeneous teachers into a unified semantic manifold.

### 4.3.2. PHASE II: FAST ADAPTER TUNING WITH SCALE-AWARE ALIGNMENT

In the second phase, we adapt the pre-trained Universal Core to a specific target MLLM (e.g., Qwen-VL) by training only the lightweight adapters ($\mathcal{A}_{in}, \mathcal{A}_{out}$). A major challenge we identified during cross-architecture transfer is Feature Scale Collapse, where the restored features typically exhibit significantly lower variance than the target HQ features, leading to activation failure in the frozen LLM. To address this, we propose Scale-Aware Alignment. Beyond the standard shape reconstruction loss $\mathcal{L}_{shape}$, which applies MSE on normalized features, we introduce a Scale Restoration Loss $\mathcal{L}_{scale}$ that explicitly enforces the energy conservation of feature distributions. Then the final objective for adapter tuning is:

$$\mathcal{L}_{adapt} = \mathcal{L}_{shape} + \alpha \mathcal{L}_{scale} \tag{5}$$

$$\mathcal{L}_{shape} = \text{MSE}(\text{Norm}(F_{restored}), \text{Norm}(F_{target})) \tag{6}$$

$$\mathcal{L}_{scale} = \| \log(\sigma(F_{restored})) - \log(\sigma(F_{target})) \|_2^2 \tag{7}$$

where $\sigma(\cdot)$ computes the standard deviation along the channel dimension. By incorporating $\mathcal{L}_{scale}$, Uni-DocRobust ensures that the injected features maintain the magnitude expected by the MLLM's projector, enabling plug-and-play integration.

### 4.3.3. PHASE III: PLUG-AND-PLAY DEPLOYMENT VIA NON-INTRUSIVE INJECTION

The final phase realizes the *train once, deploy everywhere* paradigm. Let a target MLLM consist of a visual encoder with $N$ blocks ($\mathcal{V}_{blocks}$), a modality projector/merger

($\mathcal{P}$), and a large language model ($\mathcal{M}$). The standard inference flow is $F_{vis} = \mathcal{P}(\mathcal{V}_{blocks}(I))$. To deploy Uni-DocRobust, we intercept the visual feature stream at the interface between the visual encoder and the projector. Specifically, we inject our restoration wrapper $\mathcal{W}(\cdot) = \mathcal{A}_{out}(\Phi_{core}(\mathcal{A}_{in}(\cdot)))$ precisely after the final visual block and before the projector. The modified inference flow becomes:

$$F_{robust} = \mathcal{P}\left( \mathcal{W}\left( \mathcal{V}_{blocks}(I_{LQ}) \right) + \mathcal{V}_{blocks}(I_{LQ}) \right) \tag{8}$$

This deployment strategy offers two critical advantages: (1) Zero-Cost Migration: The parameters of the host MLLM ($\mathcal{V}, \mathcal{P}, \mathcal{M}$) remain completely frozen. No gradient updates or fine-tuning are required for the host model. (2) Architecture Agnostic: The injection point of Uni-DocRobust is universally present in modern MLLMs (e.g., the merger in Qwen-VL or the MLP projector in LLaVA), making our framework compatible with various architectures via simple configuration. In practice, this allows Uni-DocRobust to be loaded as a lightweight plugin (comprising $< 5\%$ of the total parameter count) that instantly endows generic MLLMs with robustness against severe document degradations.

## 5. Experiments

### 5.1. Implementation Details

We validate the effectiveness of Uni-DocRobust and the training strategy on three open source MLLMs: LLaVA1.5-7B, Qwen3VL-2B and InternVL-3.5-2B(Chen et al., 2025). We employed up to three vision encoders during pre-training the UDRM Core: CLIP, InternViT and SigLIP. Specifically, the variant of each encoder are CLIP-ViT-Large, InternViT-300M (Chen et al., 2024) and SigLIP-So400M. In UDRM Core, the number of transformer layers is set to 6, with the feedforward dimension of the Transformer blocks being set to 2048, and the dimension of canonical space is 1024.

For model training, in the pre-training Phase I, the UDRM Core is trained for 5 epochs with a batch size of 32. We adopt an initial learning rate of 0.001, use a linear warmup over 0.5 epochs, and gradually decrease the learning rate according to a 1-cycle learning rate schedule. In the adapter tuning Phase II, the batch size is set to 32, and we only finetune the adapters for 5000 steps. All model training and inference are performed on three NVIDIA A800 GPUs.

### 5.2. Utility of DocRobust-VQA

To validate the utility of our purposed dataset DocRobust-VQA, as shown in the Table 1, we compare the performance of leading closed source, document domain-specific, and open source MLLMs on clean and corrupted images from subsets of DocRobust-VQA. Overall, all models exhibit a noticeable drop in scores on the corrupted data, which high-

| Method | Data Type | Datasets | | | |
|---|---|---|---|---|---|
| | | ChartQA | TextVQA | DocVQA | InfographicVQA |
| GPT-5 | clean | 87.2 | 79.1 | 94.5 | 80.8 |
| | corrupted | 27.42 (-59.78) | 54.12 (-24.98) | 41.55 (-52.95) | 36.82 (-43.98) |
| Gemini2.5-pro | clean | 88.9 | 80.4 | 94.7 | 81.6 |
| | corrupted | 31.25 (-57.65) | 58.62 (-21.78) | 72.04 (-22.66) | 43.15 (-38.45) |
| Claude3.5 | clean | 79.42 | 71.18 | 84.65 | 72.14 |
| | corrupted | 19.44 (-59.98) | 24.12 (-47.06) | 26.33 (-58.32) | 15.91 (-56.23) |
| TextMonkey | clean | 66.92 | 64.06 | 73.10 | 37.76 |
| | corrupted | 38.12 (-28.80) | 45.16 (-18.90) | 59.84 (-13.26) | 33.42 (-4.34) |
| TextHarmony | clean | 66.32 | 67.78 | 64.93 | 40.65 |
| | corrupted | 40.21 (-26.11) | 50.12 (-17.66) | 52.33 (-12.60) | 36.88 (-3.77) |
| mPLUG-DocOwl2.0 | clean | 69.88 | 67.11 | 80.28 | 46.70 |
| | corrupted | 34.55 (-35.33) | 49.02 (-18.09) | 62.11 (-18.17) | 35.24 (-11.46) |
| DeepSeek-VL2 | clean | 31.84 | 70.45 | 55.85 | 29.41 |
| | corrupted | 20.16 (-11.68) | 66.21 (-4.24) | 44.18 (-11.67) | 27.05 (-2.36) |
| Qwen2.5-VL-Max | clean | 88.48 | 81.46 | 95.74 | 81.84 |
| | corrupted | 56.12 (-32.36) | 64.82 (-16.64) | 83.27 (-12.47) | 61.12 (-20.72) |
| InternVL3-14B | clean | 89.6 | 82.5 | 95.8 | 82.1 |
| | corrupted | 60.4 (-29.2) | 65.2 (-17.3) | 82.6 (-13.2) | 62.8 (-19.3) |

*Table 1.* Results of leading closed source, document domain-specific, and open source MLLMs on **subsets of DocRobust-VQA**.

lights the challenging nature of our proposed DocRobust-VQA. From the perspective of model performance, the latest open-source multimodal large models have reached or even surpassed the closed-source models on OCR-related standard datasets, thanks to their well-curated document understanding data and dynamic patch processing strategies for high-resolution image inputs.

Analyzing the results across different subsets, we observe that the most significant drop in standard scores occurs in ChartQA (Masry et al., 2022), which primarily involves chart question answering. This is mainly because the distortion corruptions in DocRobust-VQA deform the image lines, resulting in the loss of critical visual information and thereby increasing the difficulty of accurately interpreting charts. In InfographicVQA (Mathew et al., 2022), we also observe a substantial decline in scores; this dataset, composed mainly of posters, often contains chart-like elements. In contrast, for DocVQA (Mathew et al., 2021) (focused on scanned documents) and TextVQA (Singh et al., 2019) (focused on scene text), the deformation has a limited impact on the textual structure. Instead, blurring and noise—which cause semantic information loss—are the main reasons for the decline in scores. These findings suggest that visual details are more crucial for chart understanding tasks, whereas semantic information plays a relatively more important role in text-based image understanding tasks.

### 5.3. Effectiveness of Uni-DocRobust

To further validate the effectiveness of our proposed Uni-DocRobust in enhancing model robustness, we conducted comprehensive experiments on both the various subsets

of DocRobust-VQA and the real-world low-quality document dataset, WildDoc. Our evaluation encompassed three diverse MLLMs: LLaVA-1.5-7B, Qwen3-VL-2B, and InternVL-3.5-2B. We compared the performance of models equipped with and without the Uni-DocRobust module, as well as against baselines directly fine-tuned via LoRA on the DocRobust-VQA training set.

The results reported in Table 2 demonstrate that integrating Uni-DocRobust significantly bolsters model capabilities. The method not only exhibits strong robustness across the synthetic test sets of DocRobust-VQA but also achieves substantial performance gains on the real-world WildDoc dataset. Remarkably, applying Uni-DocRobust alone outperforms the baseline of directly fine-tuning the original model on DocRobust-VQA using LoRA. Furthermore, combining Uni-DocRobust with LoRA fine-tuning elevates robustness to a superior level. These findings conclusively demonstrate the effectiveness of our Uni-DocRobust framework and validate that our DocRobust-VQA dataset successfully enables MLLMs to acquire robustness through training.

### 5.4. Performance on Adversarial Examples

Remarkably, we observe that even in the absence of any adversarial training or explicit defense mechanisms, training on DocRobust-VQA alone enables the Uni-DocRobust to significantly improve the model's resilience against adversarial attacks. As presented in Table 3, we evaluate adversarial robustness on four representative datasets—ChartQA, TextVQA, DocVQA, and InfographicVQA—by applying the MF-Attack (Zhao et al., 2023) method to generate adversarial samples targeting the original InternVL3.5-2B model.

| Method | Data Type | Trained Parameters | | Datasets | | | | |
|---|---|---|---|---|---|---|---|---|
| | | Uni-DocRobust | MLLM (LoRA) | ChartQA | TextVQA | DocVQA | InfographicVQA | WildDoc |
| LLaVA1.5-7B | clean | | | 55.48 | 59.56 | 75.12 | 43.61 | / |
| | corrupted | | | 26.12 | 47.05 | 64.98 | 30.22 | 28.71 |
| | | ✓ | | 42.61 | 52.33 | **70.45** | 37.19 | 34.25 |
| | | | ✓ | 41.95 | 51.78 | 69.82 | 36.55 | 33.64 |
| | | ✓ | ✓ | **44.15** | **53.82** | 70.15 | **39.24** | **35.88** |
| Qwen3VL-2B | clean | | | 72.31 | 73.18 | 91.56 | 65.42 | / |
| | corrupted | | | 45.89 | 60.12 | 60.44 | 41.87 | 32.96 |
| | | ✓ | | 56.63 | 62.48 | 61.35 | 45.12 | 38.02 |
| | | | ✓ | 55.94 | 61.85 | 60.72 | 44.45 | 37.42 |
| | | ✓ | ✓ | **58.02** | **64.15** | **62.74** | **46.88** | **39.42** |
| InternVL3.5-2B | clean | | | 77.82 | 73.44 | 86.21 | 57.32 | / |
| | corrupted | | | 35.71 | 58.11 | 73.54 | 38.92 | 32.41 |
| | | ✓ | | 59.45 | 61.87 | **77.92** | 42.53 | 38.16 |
| | | | ✓ | 58.76 | 61.22 | 77.15 | 41.88 | 37.52 |
| | | ✓ | ✓ | **61.55** | **63.88** | 77.52 | **44.62** | **40.12** |

*Table 2.* Results of Uni-DocRobust and LoRA finetuning on three open source MLLMs, tested on DocRobust-VQA and WildDoc. **Bold** and blue indicate the best and second-best results, respectively.

| Method | Datasets | | | |
|---|---|---|---|---|
| | ChartQA | TextVQA | DocVQA | InfographicVQA |
| InternVL3.5-2B | 12.72 | 15.77 | 18.31 | 9.23 |
| +Uni-DocRobust | **17.00** | **24.96** | **29.22** | **14.33** |

*Table 3.* Results on the adversarial examples generated from ChartQA, TextVQA, DocVQA and InfographicVQA datasets.

Since the attack is a white-box attack specifically targeting InternVL3.5-2B, the model exhibits a substantial performance degradation under on these adversarial examples. Nonetheless, when equipped with the DocRobust module, the model demonstrates a notable improvement in adversarial robustness, despite having never seen adversarial examples during training. This suggests that the Uni-DocRobust, through exposure to diverse degraded inputs in DocRobust-VQA, can implicitly enhance the model's robustness to perturbations beyond the training distribution.

| Method | Datasets | | | |
|---|---|---|---|---|
| | ChartQA | TextVQA | DocVQA | InfographicVQA |
| DiffBIR | 9.96 | 28.08 | 14.24 | 17.67 |
| DocRes | 28.76 | 58.11 | 70.85 | 37.09 |
| Uni-DocRobust | **59.45** | **61.87** | **77.92** | **42.53** |

*Table 4.* Results of different restoration methods applied on DocRobust-VQA, tested on InternVL3.5-2B.

## 5.5. Comparison to Pixel-level Restoration Methods

To further validate the effectiveness and superiority of our proposed feature-level restoration method, we conducted comparative experiments between Uni-DocRobust and two representative pixel-level restoration methods on the subsets of DocRobust-VQA. Specifically, we compared against DiffBIR (Lin et al., 2024), a general-purpose image restoration method, and DocRes (Zhang et al., 2024a), a method tailored for document image enhancement. Additionally,

we fine-tuned DocRes on our DocRobust-VQA dataset to ensure a fair comparison. For both pixel-level restoration methods, the evaluation pipeline is feeding the degraded images into the restoration module, and then passing the restored images to the original InternVL3.5 model for answering. The results, as shown in Table 4, demonstrate that even the fine-tuned DocRes performs significantly worse than DocRobust, with DiffBIR falling even further behind. These findings underscore the advantage of feature-level restoration over pixel-level restoration in the context of multimodal understanding under low-quality conditions, and highlight the efficacy of our proposed DocRobust.

## 5.6. Effectiveness of Multi-Teacher Distillation

To validate the effectiveness of our multi-teacher distillation strategy for the UDRM Core, we conducted ablation studies using varying numbers and types of vision encoders during Phase I training. As presented in Table 6, we evaluated the final performance on InternVL-3.5-2B after completing the full training pipeline.

First, a consistent trend is observed where increasing the number of teacher encoders generally leads to improved performance. This aligns with intuition and confirms that leveraging multiple teachers enables the UDRM Core to acquire stronger generalization capabilities. Additionally, we find that training exclusively with InternViT yields better results compared to using CLIP alone. This can be attributed to the fact that the evaluation was performed on InternVL-3.5; although the specific architectures are not identical, the results indicate that pre-training on a visual encoder from the same lineage facilitates stronger robustness enhancement when adapting Uni-DocRobust to the corresponding MLLM. These findings suggest that expanding the breadth and diversity of the teacher encoder pool is a promising avenue

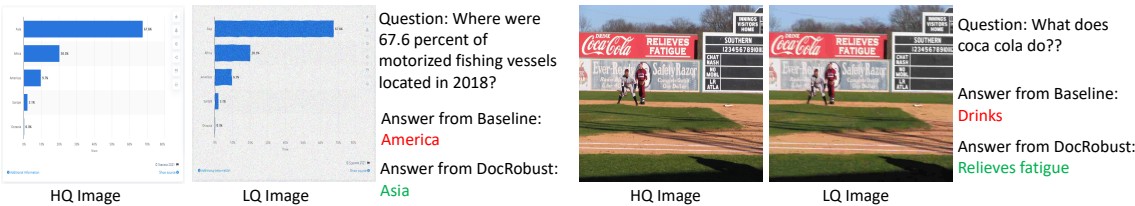

*Figure 4.* Visualization on different VQA cases.

| Method | Training Budget | Trainable Capacity | Score (Avg) |
|---|---|---|---|
| Baseline | - | 0 | 55.88 |
| Standard Adapter | Full Pretrain | ~ 1.25M | 58.36 |
| | Fast Adaptation | ~ 1.25M | 57.74 |
| Random-Initialized Uni-DocRobust | Full Pretrain | ~51.08M | 61.06 |
| | Fast Adaptation | ~51.08M | 52.29 |
| Uni-DocRobust (Ours) | Multi-Teacher Distillation + Fast Adaptation | 50.04M (Phase 1) + 1.04M (Phase 2) | **62.72** |

*Table 5.* Comparison between standard adapter and Uni-DocRobust under different training budget and capacity on InternVL-3.5-2B, tested on DocRobust-VQA.

| Vision Encoders | | | Datasets |
|---|---|---|---|
| CLIP | InternViT | SigLIP | DocRobust-VQA |
| | | | 55.88 |
| ✓ | | | 57.32 |
| | ✓ | | 58.45 |
| ✓ | ✓ | | 61.26 |
| ✓ | ✓ | ✓ | **62.72** |

*Table 6.* Results of employing different vision encoders in pretraining UDRM Core in Phase I, then deploy the Uni-DocRobust on InternVL-3.5-2B, tested on DocRobust-VQA.

for further boosting the performance of Uni-DocRobust.

Furthermore, to confirm that our performance gains originate from the architecture-agnostic restoration priors learned during Phase I rather than merely the addition of trainable parameters, we conducted a controlled ablation on InternVL3.5-2B in Table 5. We compared Uni-DocRobust against two baselines: (1) A Standard Adapter (~1.25M parameters) with zero-initialized gating, and (2) a Random-Initialized Uni-DocRobust (~51.08M parameters) trained directly on the target MLLM, bypassing Phase I distillation.

Even when the Standard Adapter is given a "Full Pretrain" budget (5 epochs), it only achieves a score of 58.36, indicating that a low-capacity target-specific module is insufficient for complex degradation recovery. More importantly, when the Random-Initialized Uni-DocRobust is given the same full training budget, it achieves 61.06—falling short of our two-stage approach (62.72). Furthermore, when restricted to the 5,000-step "Fast Adaptation" budget, the random-initialized module's performance collapses to 52.29 due to a failure to converge. This confirms that the efficacy of Uni-DocRobust relies fundamentally on the rich priors acquired during multi-teacher distillation, rather than raw parameter capacity.

### 5.7. Efficiency and Performance

As shown in Table 7, we evaluate the computational efficiency and resource costs of Uni-DocRobust against pixel-level restoration methods (DiffBIR, DocRes) and standard LoRA fine-tuning on InternVL3.5-2B. Pixel-level approaches, particularly diffusion-based models like DiffBIR, incur prohibitive inference costs (up to 563.3 GFLOPS) and fail to yield competitive VQA performance due to the semantic domain gap. While task-specific fine-tuning of DocRes improves its average score to 53.32, it remains highly inefficient, requiring 45 GPU hours. Similarly, standard LoRA fine-tuning directly on the MLLM achieves a strong score of 62.03 but demands 15 GPU hours for every new model architecture.

In contrast, Uni-DocRobust validates the efficiency of our "pre-train once, deploy everywhere" paradigm. The Phase I Universal Pre-training requires an upfront investment of 27 GPU hours to learn canonical restoration priors. However, once pre-trained, adapting the module to a new target MLLM (Phase II) requires tuning only a fraction of the parameters, taking a mere 0.3 GPU hours. This represents a 50× reduction in deployment cost compared to standard LoRA fine-tuning, while simultaneously achieving the highest average score of 62.72. Furthermore, the inference overhead added by our feature-level module is minimal, firmly establishing Uni-DocRobust as a scalable, cost-effective infrastructure for practical deployment.

| Method | Params (M) | GFLOPs | GPU hours | Avg. score |
|---|---|---|---|---|
| InternVL3.5-2B | - | - | - | 55.88 |
| + DiffBIR (w/o Finetune) | 15.8(IR)+1.6k(LDM) | - | 0 | 18.60 |
| + DocRes (w/o Finetune) | 15.2 | 563.3 | 0 | 32.01 |
| + DocRes (Finetuned) | 15.2 | 563.3 | 45 | 53.32 |
| + LoRA | 28 | 180 | 15 | 62.03 |
| **+ Uni-DocRobust (Phase I)** | 56.7 | **29.8** | 27 | / |
| **+ Uni-DocRobust (Phase I+Phase II)** | **3.2** | 180 | **0.3** | **62.72** |

*Table 7.* Average scores and resource cost comparison among Uni-DocRobust, LoRA and pixel-level restoration methods on DocRobust-VQA.

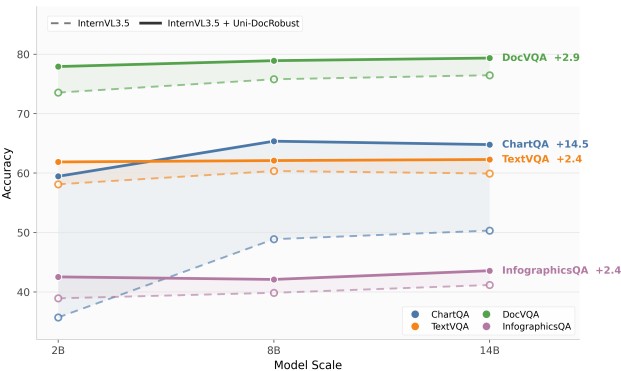

*Figure 5.* Scaling performance of Uni-Docrobust on InternVL3.5.

### 5.8. Scaling to Larger MLLM Architectures

To verify that the benefits of Uni-DocRobust scale alongside model capacity, we evaluated our framework on larger variants of the InternVL3.5 architecture (8B and 14B) using the corrupted subset of DocRobust-VQA. As shown in Fig. 5, while larger models exhibit a naturally higher baseline tolerance to degradation (e.g., the 14B baseline achieves 50.32 on ChartQA compared to the 2B baseline's 35.71), low-quality images still impose a severe performance bottleneck. Across all metrics, Uni-DocRobust provides consistent and substantial gains. On InternVL3.5-14B, our module elevated ChartQA performance from 50.32 to 64.79 and DocVQA from 76.46 to 79.35, proving that our lightweight feature restoration paradigm scales effectively to state-of-the-art, high-capacity MLLMs.

## 6. Conclusion

In this paper, we addressed the critical challenge of robust document understanding in Multi-modal Large Language Models (MLLMs), which frequently suffer from performance degradation when facing real-world image corruptions. We identified that the progress in this field has been hindered by two main bottlenecks: the scarcity of large-scale aligned training data and the lack of transferable restoration solutions. To bridge the data gap, we introduced

DocRobust-VQA, the first large-scale dataset specifically constructed for supervising robustness learning, comprising over 189K aligned image pairs. Leveraging this foundation, we proposed Uni-DocRobust, a universal plug-and-play framework that shifts the paradigm from "one-model-one-training" to "pre-train once, deploy everywhere". Our novel decoupled architecture, consisting of a Universal Restoration Core and lightweight Feature Adapters, allows for efficient cross-model migration. By implementing a structured Three-Phase Pipeline—Universal Pre-training, Fast Adapter Tuning, and Non-Intrusive Deployment—we successfully demonstrated that generic restoration primitives can be learned in a canonical space and adapted to diverse MLLMs with minimal cost. Furthermore, the proposed Scale-Aware Alignment mechanism effectively resolves the feature scale collapse issue during transfer, ensuring stable and high-performance integration. Extensive experiments show that Uni-DocRobust not only achieves state-of-the-art results on degraded document benchmarks but also serves as a cost-effective infrastructure for deploying robust MLLMs in practical applications. We believe that this work offers new insights and methodologies for robustness research in multimodal large language models.

## Acknowledgements

This work is supported by the projects of National Natural Science Foundation of China (No. 62376012), National Key R&D Program of China (2021ZD0113301), which is also a research achievement of Key Laboratory of Science, Technology and Standard in Press Industry (Key Laboratory of Intelligent Press Media Technology), and the Zhongguancun Academy (Grant No. XTS0048).

## Impact Statement

This paper presents work whose goal is to advance the field of Machine Learning, specifically focusing on the robustness of Multi-modal Large Language Models (MLLMs) in real-world document understanding. We anticipate our framework will yield several significant societal and academic impacts.

- **Accessibility and Real-World Application**

  Our framework significantly enhances the ability of MLLMs to process degraded, low-resolution, and low-quality documents. This has immediate positive implications for digitizing historical archives, extracting data from poorly scanned medical or legal records, and assisting visually impaired individuals who rely on robust visual question-answering systems to interpret physical documents in the wild.

- **Environmental Efficiency (Green AI)**

  The "plug-and-play" nature of Uni-DocRobust explicitly addresses the growing computational and environmental costs of training large models. By decoupling the Universal Restoration Core from specific visual encoders and utilizing lightweight Feature Adapters, adapting our robustness module to new state-of-the-art MLLM architectures requires only a fraction of the computational budget. This significantly lowers the barrier to entry and reduces the carbon footprint associated with heavy, full-parameter fine-tuning.

- **Addressing the Training Data Bottleneck**

  A persistent challenge in document robustness has not merely been a lack of evaluation benchmarks, but a severe scarcity of large-scale training sets. By introducing and releasing the DocRobust-VQA dataset, we provide the community with a substantial corpus of aligned clean-corrupted pairs. This resource actively removes a critical bottleneck, enabling researchers to explore and train novel robustness paradigms at scale.

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
