# OpenReview forum: "Uni-DocRobust: Universal Plug-and-Play Robustness Enhancement for Multi-modal LLMs via Feature Restoration"
_ICML.cc/2026/Conference — ICML 2026 regular_

### Official Review · Reviewer_JhVN · 2026-03-03

**Soundness:** 3
**Presentation:** 3
**Significance:** 3
**Originality:** 3
**Overall Recommendation:** 3
**Confidence:** 3

**Summary:**

The paper proposes Uni-DocRobust, a plug-and-play framework designed to enhance the robustness of Multimodal Large Language Models (MLLMs) against document image degradations. It introduces a synthetic dataset, DocRobust-VQA, and a two-stage training pipeline, where a universal restoration core" is the core components that are trained using DocRobust-VQA. Experimental results show that the proposed Uni-DocRobust works better than several baselines on typical MLLMs.

**Compliance With Llm Reviewing Policy:**

Affirmed.

**Key Questions For Authors:**

1. I'm interested to see more details about WildDoc can be provided, e.g., data size and distribution. Moreover, it would be great if more results on other real-world datasets can be provided, to justify the generalization of your proposed method.
2. Whether the benefits of the Uni-DocRobust can be maintained on larger and latest models?

**Limitations:**

I didn't see many discussion about limitation in the current version.

**Strengths And Weaknesses:**

Strengths
1. The target problem is interesting and important.
2. The proposal is reasonable and seems work on test dataset.

Major Weaknesses
1. The idea of training a universal restoration core is interesting and I agree that it can be migrated to other models, as a plug-and-play component. However, I guess adapters still need to be tuned when you applying the core to other models, which suggest that we still need to do training, not real “plug-and-play”.
2. The construction mainly relies on synthetically generated degradations (e.g., blur, noise, distortion, and compression) applied to clean document images. However, such synthetic corruptions may not fully reflect the complexity of real-world document degradations. As a result, the learned robustness may be biased toward synthetic distributions, and its generalization to real-world degraded documents remains unclear. It seems like you already know the feature of corruptions and then design an approach to address it. I highly doubt whether it aligns with the real practice.
3. Many models used in experiments are quite small, e.g., 2B, and they were released several years ago. I’m not sure whether the claimed benefits can be maintained when applying the proposal to larger and latest models.
4. It is good to see that a real-world low-quality dateset, WildDoc, is used in performance evaluation. However, there is no detail about this dataset, e.g., how many images are contained, and whether they follow similar distribution with the Uni-DocRobust. That is quite critical, because if WildDoc has similar features of distortion or blurriness, like what you did in creating Uni-DocRobust, it is not surprised that your proposal works well.

---

> ### Author Rebuttal · Authors · 2026-03-31
>
> We sincerely thank the reviewer for acknowledging the importance of our target problem and validating the effectiveness of our proposed Uni-DocRobust framework. We respond to their questions and concerns below:
>
> ## 1. Regarding the Definition of "Plug-and-Play"
>
> We completely agree with the reviewer's perspective that achieving absolute, zero-shot "Plug-and-Play" without any training remains an immense challenge. Given the significant heterogeneities in model architectures and feature dimensions among current MLLMs, a universal, zero-adaptation module is practically unattainable.
>
> In this context, Uni-DocRobust achieves a high standard of adaptability. By requiring only ~10 minutes of rapid adapter fine-tuning, it can be seamlessly integrated into various MLLMs. We considered this efficiency a close approximation of "Plug-and-Play" in the multi-modal landscape. However, we respect the reviewer's rigor and will consider refining this terminology in the revised manuscript to avoid any potential ambiguity.
>
> ## 2. Specifics of the WildDoc Dataset
>
> We thank the reviewer for pointing out the lack of detailed descriptions for the WildDoc[1] dataset. We apologize for this oversight and realize that providing specifics such as dataset size, distribution, and qualitative examples is essential.
>
> WildDoc is an open-source test dataset designed for real-world document understanding. Its source images (from DocVQA, ChartQA, TableVQA) were printed and re-captured under diverse physical environments. Crucially, WildDoc's distribution is entirely distinct from our synthetic DocRobust-VQA dataset. In fact, the "LQ Image" in Figure 1 of our manuscript is a WildDoc sample. For a comprehensive view of the image distributions, we invite the reviewer to refer to the [*link*](https://ibb.co/JWxkz0rR) and their [*huggingface page*](https://huggingface.co/datasets/ByteDance/WildDoc).
>
> ## 3. Generalization to Real-World Scenarios
>
> We fully understand the reviewer’s concern regarding whether a model trained on synthetic data can generalize to real-world degradations. We are pleased to report that `Uni-DocRobust` demonstrates robust generalization.
>
> In addition to the WildDoc results in Table 2, we have created a new real-world document dataset, **RealDoc**, featuring clean-corrupted image pairs. The RealDoc dataset was built by filtering high-quality and low-quality image pairs from two real document image datasets, Inv3dReal[2] and DocUNet[3]]. For each image pair, we used Gemini 2.5-Pro to generate QA pairs grounded in the image content, resulting in a total of 2,141 annotated QA pairs. Qualitative samples of RealDoc are available via [*link*](https://ibb.co/JWxkz0rR). We evaluated `Uni-DocRobust` (trained on DocRobust-VQA) using InternVL3.5-2B as the backbone. The results are as follows:
>
> |Model|WildDoc|RealDoc-clean|RealDoc-corrupted|
> |-|-|-|-|
> |w/o U-DR|32.41|49.34|28.89|
> |w/ U-DR|**38.16**|**52.27**|**36.76**|
>
> The results are highly encouraging: `Uni-DocRobust` significantly boosts performance on both WildDoc and RealDoc-corrupted. Notably, it even enhances performance on RealDoc-clean, suggesting that our method captures fundamental document semantics that benefit standard distributions as well.
>
> ## 4. Scaling Analysis and Model Recency
>
> We appreciate the reviewer’s query regarding model scale. While our primary experiments used 2B versions of Qwen3VL and InternVL3.5 (which are inherently more vulnerable to degradations), we have conducted scaling tests on InternVL3.5 to verify the effectiveness of our method on larger architectures. The experiments were conducted on the corrupted subset of DocRobust-VQA:
>
> |Model|ChartQA|TextVQA|DocVQA|InfographicsQA|
> |-|-|-|-|-|
> |2B|35.71|58.11|73.54|38.92|
> |2B + U-DR|**59.45**|**61.87**|**77.92**|**42.53**|
> |8B|48.87|60.34|75.78|39.85|
> |8B + U-DR|**65.36**|**62.10**|**78.91**|**42.10**|
> |14B |50.32|59.92|76.46|41.16|
> |14B + U-DR|**64.79**|**62.28**|**79.35**|**43.57**|
>
> The results indicate that even for larger MLLMs with stronger baseline robustness, low-quality images still significantly impair performance. In all cases, `Uni-DocRobust` provides consistent and substantial gains, proving its scalability.
>
> Regarding model recency, while we chose LLaVA-1.5 as a standard historical baseline, both **Qwen3VL** and **InternVL3.5** are state-of-the-art models released within the past six months. We believe that these models can represent the current frontier of the MLLM field.
>
> References:
> - [1] A. Wang, J. Tang, L. Lei, H. Feng, Q. Liu, X. Fei, J. Lu, H. Wang, W. Liu, H. Liu, Y. Liu, X. Bai and C. Huang. WildDoc: How Far Are We from Achieving Comprehensive and Robust Document Understanding in the Wild? EMNLP 2025.
> - [2] F. Hertlein, A. Naumann and P. Philipp. Inv3d: a high-resolution 3d invoice dataset for template-guided single-image document unwarping. IJDAR 2023.
> - [3] K. Ma, Z. Shu, X. Bai, J. Wang, and D. Samaras. Docunet: Document image un-warping via a stacked u-net. CVPR 2018.

---

> > ### Author Rebuttal · Reviewer_JhVN · 2026-04-03
> >
> > The added results are helpful. I'm still unclear if the same amount of parameters and training budget are given to a standard target-specific adapter/bottleneck module without the proposed design, does it achieve similar gains? Without such a control, it is unclear whether the contribution is truly the “universal restoration core,” rather than just additional trainable capacity.

---

> > > ### Author Response · Authors · 2026-04-07
> > >
> > > Thank you again for your careful reading, timely follow-up, and very helpful comments. We express our deepest gratitude to the reviewer for this rigorous and profound question. We sincerely apologize that we didn't explicitly disentangle the contribution of our `Universal Restoration Core` from the mere addition of trainable parameter capacity in the manuscript, and we completely agree that an ablation experiment of a standard adapter with the same amount of our module is required, to exhibit the contribution in our **training design**.
> > >
> > > Regarding parameter counts, `Uni-DocRobust` comprises the Stage-1 pre-trained `Core` (~50.04M) and the Stage-2 rapidly fine-tuned `Adapter In/Out` (\~0.52M each, ~1.04M total). To thoroughly address your concern and isolate the effect of our **training design**, we conducted two controlled ablation experiments on InternVL3.5-2B, using the following baselines:
> > >
> > > 1. **Standard Adapter (~1.25M):** We inserted a bottleneck MLP between the visual encoder and projector. To ensure a fair comparison aligned with modern MLLM PEFT practices, we incorporated the zero-initialized gating mechanism from LLaMA-Adapter [1]. This ensures the adapter acts as an identity mapping at initialization, preserving InternVL's pre-trained representations.
> > > 2. **Random-Initialized Uni-DocRobust (~51.08M, Random Init):** We deployed our exact full architecture (Adapters + Core) matching our proposed method's trainable capacity, but with all weights randomly initialized, **directly trained on the MLLM**, bypassing the Stage-1 distillation.
> > >
> > > Both baselines were evaluated under two training budgets: **"Full Pretrain"** (5 epochs on the DocRobust-VQA training set, mirroring Stage-1) and **"Fast Adaptation"** (5,000 steps, mirroring Stage-2). The host MLLM remained frozen throughout.
> > >
> > > The results on the corrupted DocRobust-VQA test set are presented below:
> > > |Method|Training Budget|Trainable Capacity|Score (Avg)|
> > > |:-|:-|:-:|:-:|
> > > |Baseline|-|0|55.88|
> > > |Standard Adapter|Full Pretrain|~1.25M|58.36|
> > > |Standard Adapter|Fast Adaptation|~1.25M|57.74|
> > > |Random-Initialized Uni-DocRobust|Full Pretrain|~51.08M|61.06|
> > > |Random-Initialized Uni-DocRobust|Fast Adaptation|~51.08M|52.29|
> > > |**Uni-DocRobust (Ours)**|**Multi-Teacher Distillation + Fast Adaptation**|~50.04M (Phase 1) + **1.04M (Phase 2)**| **62.72**|
> > >
> > > As the table clearly demonstrates, our proposed `Uni-DocRobust` consistently outperforms all baselines. Beyond validating the overall effectiveness of our approach, we respectfully draw the following critical conclusions from these controlled ablations:
> > >
> > > * **Standard Adapter Limitations (Row 2):** Even when a standard adapter is given the extensive "Full Pretrain" budget directly on the DocRobust-VQA dataset, it only achieves 58.36. This proves that merely injecting a standard target-specific module with limited capacity (~1.25M) is fundamentally insufficient to fully capture and resolve complex document degradation features.
> > > * **Insufficiency of Fast Adaptation without Priors (Row 3):** When the standard adapter is restricted to the "Fast Adaptation" budget (5,000 steps), its performance drops further to 57.74. It struggles to effectively learn restoration mappings within such a brief optimization window.
> > > * **Trainable Capacity vs. Learned Priors (Row 4):** This row directly answers the reviewer's **core question**. When we provide the exact same trainable capacity to the target MLLM using a randomly initialized `Uni-DocRobust` with "Full Pretrain" budget, its score (61.06) is still notably inferior to our two-stage approach (62.72). This compellingly proves that the performance leap does **not stem merely from injecting a larger parameter capacity**. Instead, it intrinsically originates from the **rich, architecture-agnostic restoration priors acquired specifically during our Stage-1 multi-teacher distillation**.
> > > * **Necessity and Efficiency of the 2-Stage Strategy (Row 5 vs. Ours):** Remarkably, when the randomly initialized 51.08M module is subjected to the "Fast Adaptation" budget, its performance collapses to 52.29. Due to the random initialization and the large parameter space, the module fails to converge within 5,000 steps, disrupting the MLLM's original pre-trained representations. In contrast, equipped with our pre-trained Core, `Uni-DocRobust` only requires tuning the lightweight ~1.04M Adapters for 5,000 steps to achieve the best results.
> > >
> > > This powerfully validates our core contribution: the decoupled training strategy (**pre-train once, deploy everywhere**) is not just efficient, but absolutely vital for achieving universal robustness in MLLMs. **We earnestly hope that these strictly controlled experiments fully resolve your core concerns. We deeply value your profound insights, and we humbly request that you might kindly consider raising the score based on these comprehensive new efforts**.
> > >
> > > [1] Zhang, R., et al. Llama-adapter: Efficient fine-tuning of language models with zero-init attention. ICLR 2024.

---

### Official Review · Reviewer_hHPf · 2026-03-10

**Soundness:** 3
**Presentation:** 3
**Significance:** 3
**Originality:** 3
**Overall Recommendation:** 4
**Confidence:** 4

**Summary:**

The paper introduces Uni-DocRobust, a plug-and-play, feature-level restoration module that enhances the robustness of multimodal LLMs (MLLMs) for document understanding under real-world degradations. The core idea is to pre-train a Universal Restoration Core in a canonical feature space via multi-teacher distillation and adapt it to any target MLLM through lightweight feature adapters, enabling non-intrusive, architecture-agnostic deployment. The authors also contribute DocRobust-VQA, a large dataset (189K HQ/LQ aligned image pairs; 417K QA pairs) tailored to robustness training and evaluation, and demonstrate consistent gains across several MLLMs and datasets, including real-world WildDoc.

**Compliance With Llm Reviewing Policy:**

Affirmed.

**Final Justification:**

I am satisfied with the authors’ detailed rebuttal, which has successfully addressed my previous concerns. Therefore, I recommend a weak accept.

**Key Questions For Authors:**

See the weaknesses.

**Limitations:**

Yes.

**Strengths And Weaknesses:**

Strengths:
1. The shift from pixel-level image restoration to a decoupled, feature-level restoration framework is highly innovative and computationally efficient. By operating in a canonical feature space, the method avoids the heavy overhead of dense pixel prediction.
2. The "pre-train once, deploy everywhere" strategy via non-intrusive injection is a major strength. The ability to freeze the host MLLM and only tune lightweight adapters (comprising less than 5% of total parameters) makes this a highly cost-effective and scalable solution.
3. The introduction of DocRobust-VQA fills a critical gap in the field. The dataset's diversity, encompassing various document domains and multi-stage corruption pipelines, provides a robust foundation for future research in degraded document understanding.
4. The authors conduct thorough experiments across multiple open-source models (LLaVA-1.5-7B, Qwen3VL-2B, InternVL-3.5-2B). The results convincingly show superiority over existing pixel-level methods like DiffBIR and DocRes.

Weaknesses:
1. When discussing generalist pixel-level restoration models, the authors ignore more advanced restoration paradigms, such as MaskDCPT [1].
2. The authors introduce a scale restoration loss L_scale to mitigate the issue of feature scale collapse. However, while the paper identifies this phenomenon and proposes a solution, the experimental section fails to provide quantitative results demonstrating the improvements yielded by this specific loss function.
3. The experimental section lacks analysis on degradation types. For example, does the method improve performance consistently across each type of degradation?
4. Could the authors add the detailed parameter settings for fine-tuning MLLMs with LoRA?

[1] Universal Image Restoration Pre-training via Masked Degradation Classification. arxiv.org/abs/2510.13282

---

> ### Author Rebuttal · Authors · 2026-03-31
>
> We deeply appreciate the reviewer's constructive feedback and the highly accurate summary of our contributions. We are greatly encouraged that the reviewer recognizes the innovative nature and computational efficiency of our decoupled, feature-level restoration approach over traditional pixel-level methods. It is particularly rewarding that our "pre-train once, deploy everywhere" strategy, the introduction of the comprehensive DocRobust-VQA dataset, and our thorough cross-architecture evaluations were viewed as major strengths.
>
> We address your specific comments and detailed concerns below.
>
> ## 1. Comparison with State-of-the-Art Pixel-Level Restoration (MaskDCPT)
>
> We sincerely thank the reviewer for pointing us to the latest state-of-the-art pixel-level restoration method, MaskDCPT. As the training scripts for MaskDCPT are currently unavailable, we utilized the pre-trained MaskDCPT model from the official repository to perform image restoration on the corrupted test set of DocRobust-VQA and the WildDoc dataset. The restored images were then fed into InternVL3.5-2B for evaluation. We compared these results with InternVL3.5-2B equipped with our Uni-DocRobust, as shown in the table below:
>
> | Model / Method | ChartQA | TextVQA | DocVQA | InfographicVQA | WildDoc |
> | :--- | :--- | :--- | :--- | :--- | :--- |
> | MaskDCPT | 27.37 | 58.92 | 44.88 | 26.16 | 30.62 |
> | **Uni-DocRobust (Ours)** | **59.45** | **61.87** | **77.92** | **42.53** | **38.16** |
>
> As the results demonstrate, the restoration performance of Uni-DocRobust comprehensively surpasses that of MaskDCPT. While MaskDCPT achieves reasonable performance on TextVQA, indicating some restorative effect, it actually degrades performance on ChartQA, DocVQA, and InfographicVQA. We attribute this to the significant domain gap between MaskDCPT's training data and document images; MaskDCPT may struggle to determine the correct restoration targets when encountering low-quality document images. This finding further highlights the critical importance of our proposed DocRobust-VQA dataset for the field of document robustness.
>
> ## 2. Quantitative Ablation Study on $L_{scale}$
>
> We greatly appreciate the reviewer for highlighting this point. In the original manuscript, we discussed the importance of $L_{scale}$ but omitted a quantitative ablation study. The following table presents the average scores on the corrupted test set of DocRobust-VQA using InternVL3.5-2B, with and without the $L_{scale}$ constraint:
>
> | Setting | Average Score |
> | :--- | :--- |
> | **w/ $L_{scale}$** | **62.72** |
> | w/o $L_{scale}$ | 24.49 |
>
> In fact, as stated in Section 4.3.2 of the main text, without $L_{scale}$, the model suffers from feature magnitude misalignment. This leads to representation collapse, making convergence extremely difficult. Furthermore, since our rapid fine-tuning in the second stage only takes 5,000 steps, fast convergence of the loss is crucial. Without $L_{scale}$, the adapter of Uni-DocRobust fails to align features properly and can even have a detrimental effect on overall performance.
>
> ## 3. Performance Across Specific Degradation Types
>
> The performance of Uni-DocRobust across specific types of degradations is indeed an aspect we had not previously evaluated. Since our DocRobust-VQA dataset was originally generated by mixing various types of degradations without explicit category labels, we constructed a new, targeted test set to evaluate our method on specific degradation types.
>
> Specifically, we focused on five categories: (1) Luminance, (2) Distortion, (3) Blurriness, (4) Noise, and (5) Compression. For each category, we randomly sampled 500 images from the broader DocRobust-VQA dataset and applied the corresponding specific degradation, resulting in a test set of 500 samples per category. We evaluated InternVL3.5-2B equipped with Uni-DocRobust on this newly constructed test set, with the results summarized below:
>
> | Model | Luminance | Distortion | Blurriness | Noise | Compression |
> | :--- | :--- | :--- | :--- | :--- | :--- |
> | InternVL3.5-2B | 57.21 | 48.53 | 43.78 | 63.29 | 65.88 |
> | **InternVL3.5-2B + Uni-DocRobust** | **68.92** | **57.33** | **50.56** | **67.15** | **69.06** |
>
> It is evident that Uni-DocRobust consistently provides significant restoration improvements across all types of document image degradations. Simultaneously, we observe that *distortion* and *blurriness* exert a more severe baseline impact on MLLMs, whereas the effects of luminance, noise, and compression are relatively smaller. We invite the reviewer to check the [*anonymous image hosting link*](https://ibb.co/JWxkz0rR) to view qualitative examples of our method's restoration capabilities across these five degradation categories.
>
> ## 4. Details for the LoRA Fine-tuning Baselines
> Due to the strict word limit of this rebuttal, we kindly invite the reviewer to refer to our **Response 3.** to **Reviewer L5jJ**, where we have addressed this exact same question in detail.

---

> > ### Author Rebuttal · Reviewer_hHPf · 2026-04-02
> >
> > I appreciate the authors’ detailed rebuttal. They have sufficiently addressed my previous concerns, and the clarifications are clear and convincing. Therefore, I decide to keep my original rating.

---

> > > ### Author Response · Authors · 2026-04-07
> > >
> > > We respectfully thank the reviewer for the thoughtful confirmation and for maintaining the positive rating. Your rigorous initial feedback was instrumental in helping us strengthen the clarity and quality of our manuscript. Thank you once again for your invaluable time and support!

---

### Official Review · Reviewer_Gf71 · 2026-03-12

**Soundness:** 2
**Presentation:** 3
**Significance:** 2
**Originality:** 3
**Overall Recommendation:** 4
**Confidence:** 2

**Summary:**

The manuscript proposes "Uni-DocRobust", which is a framework intended to improve the robustness of Multi-modal Large Language Models (MLLMs) when they process degraded document images. The authors first introduce a new dataset named DocRobust-VQA, which includes 189K pairs of aligned low-quality and high-quality document images, alongside 417K question-answering pairs. Furthermore, they propose a feature-level restoration module that operates in a decoupled architecture. This architecture utilizes a frozen Universal Restoration Core, which is trained using multi-teacher distillation, and connects to the target MLLM via lightweight Feature Adapters. This allows for a plug-and-play integration without the necessity of fine-tuning the original MLLM weights.

**Compliance With Llm Reviewing Policy:**

Affirmed.

**Final Justification:**

Please refer to Rebuttal Acknowledgment.

**Key Questions For Authors:**

- Could you please elucidate the absolute necessity of the multi-teacher distillation phase? The data suggests that employing an encoder from the same lineage (e.g., InternViT for InternVL-3.5) yields stronger robustness enhancement. Does this not negate the benefit of the "universal" canonical space?
- How does the Universal Restoration Core perform when subjected to real-world image degradations that fall completely outside the five synthetic corruption categories (Luminance, Distortion, Blurriness, Noise, and Compression) utilized during the construction of the DocRobust-VQA dataset?
- What is the precise computational overhead (in GPU hours and memory) required for Phase I (Universal Pre-training via Multi-Teacher Distillation) when distilling from multiple large models like CLIP, InternViT, and SigLIP simultaneously?

**Limitations:**

Yes

**Strengths And Weaknesses:**

Strengths:
- The conceptualization of decoupling the restoration function from specific visual encoders into a plug-and-play feature-level module is a novel combination of existing adapter methodologies.
- Addressing the performance degradation of MLLMs on low-quality documents (such as those with noise, blur, or low resolution) is a highly relevant problem for real-world applications.
- The non-intrusive injection strategy is commendable, as it permits zero-cost migration where the parameters of the host MLLM remain completely frozen.

Weights:
- The core premise of the "universal" multi-teacher distillation is somewhat contradicted by the authors' own ablation studies. The manuscript states that training exclusively with the InternViT encoder yields better results than using CLIP alone when evaluating on the InternVL-3.5 model. This indicates that pre-training on a visual encoder of the same lineage facilitates stronger robustness enhancement. Therefore, the necessity of the computationally expensive multi-teacher distillation over a simple, lineage-specific adapter is not strictly proven.
- The comparative analysis against baseline methods is somewhat limited in scope. The authors primarily compare their feature-level restoration method against pixel-level restoration methods, specifically DiffBIR and DocRes. A comparison with other contemporary feature-level robustness techniques or advanced prompt-engineering defenses is lacking, which limits the assessment of the proposed method's significance within the broader MLLM robustness landscape.

---

> ### Author Rebuttal · Authors · 2026-03-31
>
> We sincerely thank the reviewer for their precise summary and for recognizing the high real-world relevance of our target problem, as well as the novelty and practical value of our non-intrusive, plug-and-play feature-level restoration framework.
>
> We respond to their questions and concerns below:
>
> ## 1. The Necessity of Multi-Teacher Distillation
>
> We fully understand and respect the reviewer's question arising from the experimental results. While pre-training the Uni-DocRobust Core on a target MLLM's specific visual encoder yields better performance than a mismatched encoder, the table below demonstrates why our multi-teacher distillation is crucial:
>
> |Pre-training Setting | LLaVA1.5-7B | Qwen3VL-2B | InternVL3.5-2B |
> |-|-|-|-|
> |Baseline|46.13|54.24|55.88|
> |CLIP|49.94|55.76|57.32|
> |SigLIP|47.86|56.38|57.98|
> |InternViT|47.52|54.44|58.45|
> |**3-Teacher Distillation** |**53.58**|**57.62**|**62.72**|
>
> As the results clearly demonstrate:
> 1. While a Core pre-trained on a matched visual encoder performs reasonably well on its corresponding MLLM, its performance deteriorates significantly when deployed to *other* MLLM architectures.
> 2. Even when compared to the best single-encoder pre-training (the matched encoder), our multi-teacher distillation approach achieves strictly superior performance across all settings.
>
> Therefore, we wish to clarify the indispensable value of multi-teacher distillation in two aspects:
> First, it substantially elevates the upper bound of restoration performance compared to single-encoder pre-training. Second, the distilled `Uni-DocRobust Core` can be universally deployed to *any* MLLM via a rapid, lightweight second-stage adaptation. This eliminates the massive overhead of retraining a separate Core for every new MLLM architecture.
>
> To further substantiate this "universal" capability, we conducted another ablation: pre-training the Core jointly on CLIP and SigLIP (completely excluding InternViT), and testing its adaptation on InternVL:
>
> |Pre-training Setting|Performance on InternVL|
> |-|-|
> |CLIP|57.32|
> |SigLIP|57.98|
> |CLIP + SigLIP| **58.65**|
> |InternViT|58.45|
>
> We are thrilled to observe that even without exposing the Core to the target MLLM's specific visual architecture during pre-training, joint training on diverse encoders can yield better generalization than training solely on the matched target encoder. This strongly validates the **universal** ability of our method.
>
> ## 2. Clarification on Feature-Level Baselines
>
> We are grateful for the reviewer’s constructive feedback regarding the lack of comparisons with other feature-level methods. To the best of our knowledge, `Uni-DocRobust` is the *first* work to propose a plug-and-play feature-level restoration specifically designed for MLLMs.
>
> Our feature-level restoration aims to mirror direct image-level restoration, characterized by the ability to be used on *any* MLLM. We achieved this through our universal Core and rapid adaptation, which is why we primarily benchmarked against image restoration methods. In contrast, other contemporary "feature-level" robustness methods generally do not reconstruct or restore degraded features; rather, they require invasive direct training of the MLLM parameters themselves to acquire robustness, thus lacking our lightweight, rapid-adaptation property.
>
> ## 3. Comparison with Prompt Engineering
>
> Following the reviewer's suggestion to explore Prompt Engineering, we conducted two experiments on InternVL3.5-2B:
> 1.  Robust-caption prompting: Prepending *"Please note that the image may contain degradations such as noise, blur, and distortion. Carefully examine the text before answering."*
> 2.  CoT-prompting: Appending *"Let's think step by step."*
>
> The results are as follows:
>
> |Strategy|Score|
> |-|-|
> |Baseline|55.88|
> |Robust-caption prompting|53.14|
> |CoT-prompting|32.36|
> |Uni-DocRobust|62.72|
>
> Surprisingly, these prompts degraded performance. CoT caused a catastrophic drop because our evaluated VQA datasets strictly require precise, short-phrase outputs. MLLMs are pre-trained to follow this exact format, however, altering prompts induces verbose reasoning; even if the answer is correct, strict exact-match metrics severely penalize these responses. This elegantly highlights a core advantage of Uni-DocRobust: it operates invisibly at the perceptual level, requiring zero changes to the model's parameters, instruction formats, or evaluation pipelines.
>
> ## 4. Details on Stage-1 Training Cost
>
> The first stage was trained on our 417,000-sample training set for 5 epochs. The entire process was completed on 3 NVIDIA L40S GPUs. The multi-teacher distillation across the three visual encoders took a total of 13.1 hours (i.e., 39.3 GPU hours) and 21Gb memory on each GPU.
>
> ## 5. Specifics of the WildDoc Dataset & Generalization to Real-World Scenarios
>
> Due to the word limit of this rebuttal, we invite the reviewer to refer to our **Response 2. & 3.** to **Reviewer JhVN**, where we have addressed this question in detail.

---

> > ### Author Rebuttal · Reviewer_Gf71 · 2026-04-02
> >
> > The ablation study about CLIP and SigLIP testing on InternVL clarifies the necessity of multi-teacher distillation for cross-architecture generalization. The prompt engineering experiment proves feature-level restoration is more practical than modifying text instructions, since CoT causes strict metric penalties. Although the comparison with other feature-level works remains slightly thin, the provided evidences are acceptable. I upgrade the score to Weak Accept (leaning towards borderline).

---

> > > ### Author Response · Authors · 2026-04-07
> > >
> > > We sincerely thank the reviewer for the thoughtful reassessment and for upgrading the score to a Weak Accept. We are thrilled that our additional ablation studies successfully clarified the necessity of multi-teacher distillation for cross-architecture generalization. We are also glad that the prompt engineering experiments effectively demonstrated the practical advantages of our approach. We deeply appreciate your rigorous guidance and continued support!

---

### Official Review · Reviewer_L5jJ · 2026-03-12

**Soundness:** 3
**Presentation:** 3
**Significance:** 3
**Originality:** 3
**Overall Recommendation:** 4
**Confidence:** 3

**Summary:**

The paper introduces Uni-DocRobust, a plug-and-play, feature-level restoration module that enhances the robustness of multimodal LLMs (MLLMs) for document understanding under real-world degradations. The core idea is to pre-train a Universal Restoration Core in a canonical feature space via multi-teacher distillation and adapt it to any target MLLM through lightweight feature adapters, enabling non-intrusive, architecture-agnostic deployment. The authors also contribute DocRobust-VQA, a large dataset (189K HQ/LQ aligned image pairs; 417K QA pairs) tailored to robustness training and evaluation, and demonstrate consistent gains across several benchmarks.

**Compliance With Llm Reviewing Policy:**

Affirmed.

**Final Justification:**

The authors address my main concerns.

**Key Questions For Authors:**

please see weaknesses.

**Limitations:**

yes

**Strengths And Weaknesses:**

Strengths
- The feature-level restoration approach decoupled from specific vision encoders is a fresh and compelling alternative to pixel-level restoration for MLLMs, avoiding heavy image-domain computation and overfitting to perceptual metrics.
- The evaluation covers major benchmarks like DocVQA and ChartQA, and further proves effectiveness on the real-world WildDoc dataset, demonstrating generalization across both synthetic and real degradations.

Weakbesses
- The paper does not report performance on high-quality (clean) images when the module is enabled. Robustness enhancements often lead to a "robustness tax" (accuracy drop on clean inputs), and the lack of this data leaves a gap in understanding its real-world trade-offs.
- Details for the LoRA fine-tuning baselines (e.g., training data volume, iterations) are insufficient to determine if the comparison was conducted under a similar training budget. Comparison with recent Test-Time Adaptation (TTA) methods is also missing.

---

> ### Author Rebuttal · Authors · 2026-03-31
>
> We deeply appreciate the reviewer's constructive feedback and the precise summary of our contributions. We are greatly encouraged that the reviewer recognizes the value of our plug-and-play, feature-level restoration paradigm as a compelling alternative to pixel-level methods. We are also glad that our comprehensive evaluation—demonstrating strong generalization across both synthetic and real-world degradations (e.g., WildDoc)—resonated well with the reviewer.
>
> We respond to their questions and concerns below:
>
> ## 1. Performance on Clean Data & The Trade-off Concern
>
> We sincerely thank the reviewer for raising this insightful point. Indeed, in the context of enhancing MLLM robustness, parameter modifications often incur a trade-off, where specific robustness gains might lead to performance degradation in general/clean scenarios. To address this concern, we present the performance of MLLMs equipped with Uni-DocRobust on the clean subset of the DocRobust-VQA dataset in the table below:
> | | ChartQA | TextVQA | DocVQA | InfographicsVQA |
> |-|-|-|-|-|
> | LLaVA1.5-7B | 55.48   | 59.56   | 75.12  | 43.61 |
> | LLaVA1.5-7B + Uni-DocRobust    | 54.79   | 59.26 | 74.98  | 44.24|
> | Qwen3VL-2B | 72.31   | 73.18   | 91.56  | 65.42|
> | Qwen3VL-2B + Uni-DocRobust | 72.48   | 72.65 | 90.28  | 65.89|
> | InternVL3.5-2B | 77.82 | 73.44 | 86.21  | 57.32 |
> | InternVL3.5-2B + Uni-DocRobust | 77.14   | 72.78   | 86.33  | 58.45 |
>
> As shown, the integration of Uni-DocRobust into LLaVA-7B, Qwen3VL-2B, and InternVL3.5-2B has a negligible impact on their performance on clean data. Any observed performance drops are marginal, and notably, the performance even improves on certain datasets. This demonstrates that training on the DocRobust-VQA dataset is **almost harmless to normal data performance** and may **even enhance the model's performance on hard examples** within the original clean distribution.
>
> ## 2. Comparison with Test-Time Adaptation (TTA) Methods
>
> We highly appreciate the reviewer for pointing out the lack of comparisons in this area. To our knowledge, most traditional TTA methods (e.g., [1, 2]) aim to achieve generalization on zero-shot or out-of-distribution data during inference. However, many of these methods are primarily designed for Vision-Language Models (VLMs) rather than MLLMs. Furthermore, they typically require individual, model-specific training pipelines, **lacking the versatile, rapid-deployment capability of Uni-DocRobust across diverse MLLM architectures**. Consequently, our original submission primarily focused on comparisons with general image restoration methods.
>
> Nevertheless, we identified a state-of-the-art TTRV approach [3] that employs test-time reinforcement learning to enhance model generalization. Assuming this aligns with the reviewer's suggestion, we conducted an additional comparative experiment on InternVL3.5-2B , evaluating both methods on the corrupted DocRobust-VQA and WildDoc datasets. The results are summarized below:
>
> | | ChartQA | TextVQA | DocVQA | InfographicsVQA | WildDoc | Inference time |
> | - | - | - | - |- |- |- |
> | TTRV          | 38.67   | 57.47   | 74.28  | 40.54   | 31.20   | ~8s   |
> | Uni-DocRobust | 59.45   | 61.87   | 77.92  | 42.53  | 38.16   | ~1s |
>
> As the results indicate, the TTRV method proposed in [3] not only yields inferior restoration performance on low-quality document images compared to Uni-DocRobust, but it also requires significantly longer inference time due to test-time optimization. This comparison further highlights **the superior efficiency and effectiveness of our "plug-and-play" paradigm**.
>
> ## 3. Details for the LoRA Fine-tuning Baselines
>
> Regarding the LoRA hyperparameter settings, we followed the fine-tuning configurations provided in the official code repositories for each respective MLLM:
> - For LLaVA-7B: $r=128$, $\alpha=256$, learning rate $= 2e-4$, batch size $= 16$.
> - For InternVL3.5-2B: $r=128$, $\alpha=256$, learning rate $= 4e-5$, batch size $= 16$.
> - For Qwen3.5-2B: $r=64$, $\alpha=128$, learning rate $= 1e-5$, batch size $= 4$.
>
> All models were trained for 1 epoch on the DocRobust-VQA training set. Specifically, for the experiments utilizing both Uni-DocRobust and LoRA (as shown in Table 2), our training pipeline is as follows: we first perform a rapid fine-tuning of Uni-DocRobust on a small subset of 5,000 samples. Subsequently, we deploy the module into the model and jointly train it during the LoRA fine-tuning phase, keeping the parameters of Uni-DocRobust trainable.
>
> References:
>
> - [1] Ebrahimi, S., Arik, S.Ö., & Pfister, T. Test-Time Adaptation for Visual Document Understanding. ArXiv, abs/2206.07240.
> - [2] A. Karmanov, D. Guan, S. Lu, A. El Saddik and E. Xin. Efficient Test-Time Adaptation of Vision-Language Models. CVPR 2024.
> - [3] A. Singh, S. Marjit, W. Lin, P. Gavrikov, S. Yeung-Levy, H. Kuehne, R. Feris, S. Doveh, J. Glass and M. Jehanzeb Mirza. TTRV: Test-Time Reinforcement Learning for Vision Language Models. CVPR 2026.

---

> > ### Author Rebuttal · Reviewer_L5jJ · 2026-04-06
> >
> > Thank you for the response. I will maintain the positive score.

---

> > > ### Author Response · Authors · 2026-04-07
> > >
> > > We sincerely thank the reviewer for taking the time to read our rebuttal and for maintaining the positive score. Your insightful and constructive feedback has been absolutely invaluable in helping us strengthen the quality of our manuscript. Thank you once again for your continued support and guidance throughout the review process!

---

### Decision · Program_Chairs · 2026-04-30

**Decision:**

Accept (regular)

**Comment:**

This paper proposes Uni-DocRobust, a feature-level robustness enhancement framework for multimodal LLMs in degraded document understanding, together with the DocRobust-VQA dataset of aligned clean/degraded document image pairs and QA annotations. Overall, the reviewers agree that the paper investigates an important and practically meaningful problem, and several reviewers found the feature-level restoration design, cross-architecture deployment perspective, and dataset contribution promising. The main strengths of the paper lie in addressing a relevant robustness problem for document MLLMs, introducing a reasonably novel feature-level restoration perspective, and providing empirical validation across multiple models and datasets. While some reservations remain after rebuttal, including the somewhat strong framing of the method as a “universal plug-and-play” and “pre-train once, deploy everywhere” solution, the still not fully closed attribution question regarding the source of the gains, and the relatively limited comparison to broader feature-level robustness methods (JhVN, Gf71), the AC finds that the rebuttal provided substantial and helpful additional evidence on the main concerns. On balance, the overall technical merit and practical value of the paper are sufficient to support a weak accept recommendation.